# WEAK CONTRACTION MAPPING AND OPTIMIZATION / CONFERENCE SUBMISSIONS

## ABSTRACT

The weak contraction mapping is a self mapping that the range is always a subset of the domain, which admits a unique fixed-point. The iteration of weak contraction mapping is a Cauchy sequence that yields the unique fixed-point. A gradient-free optimization method as an application of weak contraction mapping is proposed to achieve global minimum convergence. The optimization method is robust to local minima and initial point position.

## 1 INTRODUCTION

Many gradient-based optimization methods, such as gradient descent method, Newton's method and so on, face great challenges in finding the global minimum point of a function. As is known, searching for the global minimum of a function with many local minima is difficult. In principle, the information from the derivative of a single point is not sufficient for us to know the global geometry property of the function. For a successful minimum point convergence, the initial point is required to be sufficiently good and the derivative calculation need to be accurate enough. In the gradient-based methods, the domain of searching area will be divided into several subsets with regards to local minima. And eventually it will converge to one local minimum depends on where the initial point locates at.

Let (X,d) be a metric space and let T:X → X be a mapping. For the inequality that,

$$d(T(x), T(y)) \le qd(x, y), \forall x, y \in X. \tag{1}$$

if $q \in [0, 1)$, T is called contractive; if $q \in [0, 1]$, T is called nonexpansive; if $q < \infty$, T is called Lipschitz continuous(1; 2). The gradient-based methods are usually nonexpansive mapping the solution exists but is not unique for general situation. For instance, if the gradient descent method is written as a mapping T and the objective function has many local minima, then there are many fixed points accordingly. From the perspective of spectra of bounded operator, for a nonexpansive mapping any minima of the objective function is an eigenvector of eigenvalue equation $T(x) = \lambda x$ ,in which $\lambda = 1$. In the optimization problem, nonexpansive mapping sometimes works but their disadvantages are obvious. Because both the existence and uniqueness of solution are important so that the contractive mapping is more favored than the nonexpansive mapping(3; 4).

Banach fixed-point theorem is a very powerful method to solve linear or nonlinear system. But for optimization problems, the condition of contraction mapping $T : X \to X$ that $d(T(x), T(y)) \le qd(x, y)$ is usually too strict and luxury. In the paper, we are trying to extend the Banach fixed-point theorem to an applicable method for optimization problem, which is called weak contraction mapping.

In short, weak contraction mapping is a self mapping that always map to the subset of its domain. It is proven that weak contraction mapping admits a fixed-point in the following section. How to apply the weak contraction mapping to solve an optimization problem? Geometrically, given a point, we calculate the height of this point and utilize a hyperplane at the same height to cut the objective function, where the intersection between the hyperplane and the objective function will form a contour or contours. And then map to a point insider a contour, which the range of this mapping is always the subset of its domain. The iteration of the weak contraction mapping yields a fixed-point, which coincides with the global minimum of the objective function.

## 2 WEAK CONTRACTION MAPPING AND THE FIXED-POINT

In this section, the concept of weak contraction mapping and its fixed-point will be discussed in detail.

**Definition 1.** *Let $(X, d \ and \ D)$ be a metric space. Both the metric measurement d and D are defined in the space. And the metric measurement $D(X)$ refers to the maximum distance between two points in the vector space X:*

$$D(X) := sup\{d(x, y), \forall x, y \in X\} \tag{2}$$

**Definition 2.** *Let $(X, d \ and \ D)$ be a complete metric space. Then a mapping $T : X \to X$ is called weak contraction mapping on X if there exists $X_0 \subseteq X$ that $D(X_0) < \infty$, $X_{i+1} = \mathcal{R}(T(X_i)) \subset X_i, \forall i \in \mathbb{N}_0$ and a $q \in [0, 1)$ such that $D(X_{i+1}) \leq qD(X_i)$.*

The weak contraction mapping is an extension of contraction map with a looser requirement that $D(X_{i+1}) \leq q(X_i), X_{i+1} \subset X_i, T(x_i) \in X_{i+1} \ and \ x_i \in X_i$.

**Theorem 1.** *Let $(X, d \ and \ D)$ be a non-empty complete metric space with weak contraction mapping $T : X \to X$. Then T admits a unique fixed-point $x^*$ in X when $X_0$ is decided.*

Let $x_0 \in X$ be arbitrary and define a sequence $\{x_n\}$ be setting: $x_n = T(x_{n-1})$. The Theorem.1 is proven in the following lemmas.

By definition, there exists $q \in [0, 1)$ such that $D(X_{i+1}) \leq qD(X_i), \forall X_i$ indicate the $X_1$ must be bounded such that $D(X_1) < \infty$.

**Lemma 1.1.** *$\{x_n\}$ is a Cauchy sequence in $(X, d \ and \ D)$ and hence converges to a limit $x^*$ in $X_0$.*

Proof.Let $m, n \in \mathbb{N}$ such that $m > n$.

$$d(x_m, x_n) \leq D(X_n)$$
$$\leq q^n D(X_0)$$

Let $\epsilon > 0$ be arbitrary, since $q \in [0, 1)$, we can find a large $N \in \mathbb{N}$ such that

$$q^N \leq \frac{\epsilon}{D(X_0)}.$$

Hence, by choosing $m, n$ large enough:

$$d(x_m, x_n) \leq q^n D(X_0) \leq \frac{\epsilon}{D(X_0)} D(X_0) = \epsilon.$$

Thus, $\{x_n\}$ is Cauchy and converges to a point $x^* \in X_0$. □

**Lemma 1.2.** *$x^*$ is a fixed-point of T in $X_0$.*

Proof.

$$\lim_{x \to \infty} x_n = \lim_{n \to \infty} T(x_{n-1})$$
$$\lim_{x \to \infty} x_n = T(\lim_{n \to \infty} x_{n-1})$$

Thus,$x^* = T(x^*)$.(6) □

**Lemma 1.3.**

$$\lim_{i \to \infty} D(X_i) = 0$$

Proof.

$$\lim_{i \to \infty} D(X_i) \leq \lim_{i \to \infty} q^n D(X_0) = 0$$

**Lemma 1.4.** *$x^*$ is the only fixed-point of T in $(X, d)$ with regards to a specific $X_0$.*

Proof. Suppose there exists another fixed-point y that $T(y) = y$, then choose the subspace $X_i$ that both the $x^*$ and $y$ are the only elements in $X_i$. By definition, $X_{i+1} = \mathcal{R}(T(X_i))$ so that, both the $x^*$ and $y$ are elements in $X_{i+1}$, namely,

$$0 \leq d(x^*, y) \leq D(X_{i+1}) \leq qD(X_i) = qd(x^*, y)$$
$$d(x^*, y) = 0$$

Thus $x^* = y$.(6) $\square$

Let a hyperplane L cut the objective function f(x), the intersection of L and f(x) forms a contour (or contours). Observing that the contour (or contours) will divide X into two subspaces the higher subspace $X^> := \{x \mid f(x) > h, \forall x \in X\}$ and the lower subspace $X^{\leq} := \{x \mid f(x) \leq h, \forall x \in X\}$. The map $T : X \to X$ that $x_{i+1} = T(x_i) \in X^{\leq}$ and there exists $q \in [0, 1)$ such that $D(X_{i+1}^{\leq}) \leq qD(X_i^{\leq})$.

Geometrically, the range of weak contraction mapping shrinks over iterates, such that, $X \supseteq X_0^{\leq} \supset X_1^{\leq} \supset \cdots \supset X_i^{\leq}$. Based on lemma.1.3, the $D(X_i)$ measurement converges to zero as i goes to infinity, namely,

$$\lim_{i \to \infty} D(X_i) = 0$$

And the sequence of iteration $x_{i+1} = Tx_i$ that $x_0$, $x_1 = Tx_0$, ...,$x_{i+1} = T^i x_0$ is Cauchy sequence that converge to the global minimum of objective function f(x) if the f(x) has a unique global minimum point.

**Lemma 1.5.** *Provided there is a unique global minimum point of an objective function, then $x^*$ is the global minimum point of the function.*

Proof. The global minimum point must be insider the lower space $\{X_i^{\leq}, \forall i \in \mathbb{N}_0\}$. Similar to the proof of uniqueness of fixed-point, suppose the global minimum point $x_{min}$ of objective function is different from $x^*$. By measuring the distance between fixed-point $X^*$ and the global minimum point $x_{min}$,

$$0 \leq d(x^*, x_{min}) \leq \lim_{i \to \infty} D(X_i^{\leq}) = 0$$

The inequality above indicates $d(x^*, x_{min}) = 0$, thus $x^* = x_{min}$. $\square$

Compared with contraction map, the weak contraction map is much easier to implement in the optimization problem as the requirement $D(T(x_i)) \leq D(x_i)$ is looser than $d(T(x), T(y)) \leq d(x, y)$. Different from $d(T(x), T(y)) \leq d(x, y)$, the inequality $D(T(x_i)) \leq D(x_i)$ doesn't require $x_i$ in sequence $\{x_n\}$ must move closer to each other for every step but confine the range of $x_i$ to be smaller and smaller. Therefore, the sequence $\{x_n\}$ can still be a Cauchy and has the asymptotic behavior to converge to the fixed-point.

## 3 OPTIMIZATION ALGORITHM IMPLEMENTATION

Given the objective function $f(x)$ has a unique global minimum point, the task is to find a weak contraction mapping $T : X \to X$ such that the unique fixed-point of mapping is the global minimum point of the function. The weak contraction map for the optimization problem can be implemented in following way. First, provide one arbitrary initial point $x_0$ to the function and calculate the height $L = f(x_0)$ of the point and this height is the corresponding contours' level; Second, given the initial point map to another point inside the contour. One practical way is to solve the equation $f(x) = L$ and get $n$ number of roots which locate on a contour(or contours) and then the average of these roots is the updated searching point. And then repeat these process until the iteration of searching point converge.

This contour-based optimization algorithm utilizes the root-finding algorithm to solve the equation $f(x) = L$ and get $n$ number of roots. The starting point for the root-finding algorithm is generated by a random number generator. This stochastic process will help the roots to some extent widely distribute over the contour rather than concentrate on somewhere.

The inequality $d(x_m, x_n) \leq q^n D(X_0)$ indicates the rate of convergence, namely, the smaller q is the high rate of convergence will be achieved. Geometrically, the equation $f(x) = L$ is the intersection of the objective function and a hyperplane whose height is $L$. We hope the hyperplane move downward in a big step during each iterate and the centroid of the contour refers to the most likely minimum position. Therefore, averaging the roots as an easy and effective way to map somewhere near the centroid. And there is a trade-off between the number of roots on the contour and the rate of convergence. The larger amount of roots on the contour, the more likely the average locates closer to the centroid of the contour, and then the less iterates are required for convergence. In another words, the more time spend on finding roots on a contour, the less time spend on the iteration, vice verse.

The global minimum point $x^*$ is the fixed-point of the iteration $x_{i+1} = Tx_i$ and solves the equation $T(x^*) = x^*$.(8) And based on the above theorem, the sequence $x, T(x), T^2(x), T^3(x)...T^n(x)...$ is Cauchy that converge to the fixed-point.

First of all, the optimization algorithm has been tested on a convex function the Sphere function $f(x) = \sum x_i^2$. The minimum is (0,0,0), where $f(0,0,0) = 0$. The iterations of roots and contours is shown in FIG.1 and the update of searching point is shown in TABLE.1.

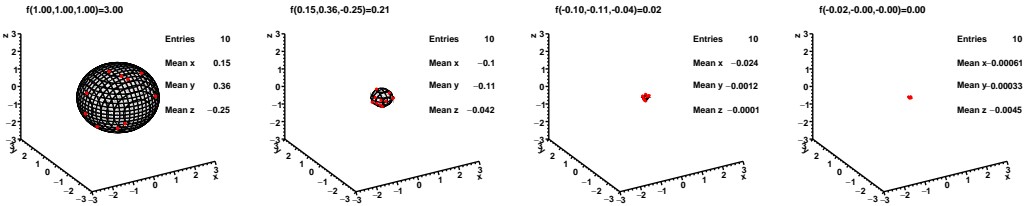

Figure 1: The red point markers are the roots and spherical surface is the contour is 3D space for each iteration.

| iteration | average of roots | level of contour |
|---|---|---|
| 0 | (1.00,1.00,1.00) | 3.0000 |
| 1 | (0.15,0.36,-0.25) | 0.2146 |
| 2 | (-0.1,-0.11,-0.042) | 0.0237 |
| 3 | (-0.024,-0.0012,-0.0001) | 0.0004 |
| 4 | (-0.0061,-0.00033,-0.0045) | |

Table 1: When the optimization method is tested on Sphere function, the average of roots and the level of contour for each iteration is shown above.

Furthermore, we test the optimization algorithm on McCormick function. And the first 4 iteration of roots and contour is shown in FIG.2 and the detailed iteration of searching point from the numerical calculation is shown in TABLE.2. The test result indicate the average of roots can move towards the global minimum point (-0.54719,-1.54719), where $f(-0.54719, -1.54719) = -1.9133$.

It is worth noting that the size of contour become smaller and smaller during the iterative process and eventually converge to a point, which is the minimum point of the function.

## 4 CONVEX SUBSETS DECOMPOSITION

As shown in the previous examples, averaging the roots on the contour is an effective approach to map a point inside the interior of the lower space $X_\leq$ when it is convex. However, in general situation, the space $X_\leq$ is not guaranteed to be convex. In that case, it is important to decompose the lower space $X_\leq$ into several convex subsets.

In this study, the key intermediate step is to check whether two roots belong to the same convex subset and decompose all roots into several convex subsets accordingly. One practical way to achieve that is to pair each two roots and scan function's value along the segment between the two roots

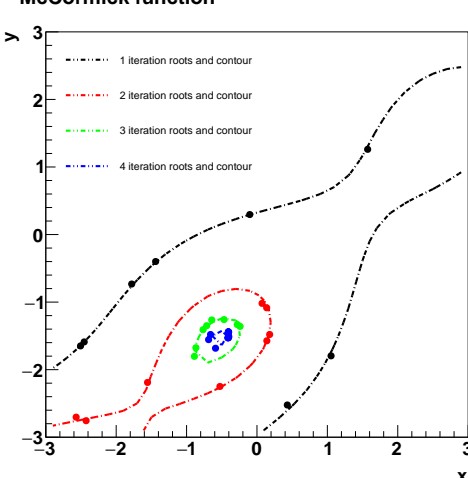

Figure 2: The first 4 iteration results are drawn to illustrate the procedure of optimization test on the McCormick function.

| iteration | average of roots | level of contour |
|---|---|---|
| 0 | (2.00000,2.0000) | 2.2431975047 |
| 1 | (-0.6409,-0.8826) | -1.1857067055 |
| 2 | (-0.8073,-1.8803) | -1.7770492114 |
| 3 | (-0.5962,-1.4248) | -1.8814760998 |
| 4 | (-0.4785,-1.5162) | -1.9074191216 |
| 5 | (-0.5640,-1.5686) | -1.9125755974 |
| 6 | (-0.5561,-1.5467) | -1.9131043354 |
| 7 | (-0.5474,-1.5465) | -1.9132219834 |
| 8 | (-0.5473,-1.5472) | |

Table 2: When the optimization method is tested on McCormick function, the average of roots and the level of contour for each iteration is shown above.

and check whether there exists a point higher than contour's level. Loosely speaking, if two roots belong to the same convex subset, the value of function along the segment is always lower than the contour's level. Otherwise, the value of function at somewhere along the segment will be higher than the contour's level. Traverse all the roots and apply this examination on them, then we can decompose the roots with regards to different convex subsets. This method is important to map a point insider interior of a contour and make hyperplane move downwards.

To check whether two roots belong to the same convex subset, $N$ number of random points along the segment between two roots are checked whether higher than the contour's level or not. When we want to check the function's value along the segment between $r_m$ and $r_n$. The vector $\vec{k} = \vec{r}_m - \vec{r}_n$ is calculated so that the random point $p_i$ locate on the segment can be written as $\vec{p_i} = \vec{r}_n + \epsilon(\vec{r}_m - \vec{r}_n), \epsilon \in (0,1)$, where the $\epsilon$ is a uniform random number from 0 to 1. Then check whether the inequality holds for all random point such that $f(p_i) < f(r_m), \forall i \leq N$. Obviously, the more random points on the segment are checked, the less likely the point higher than contour's level is missed(9; 10).

After the set of roots are decomposed into several convex subsets, the averages of roots with regards to each subsets are calculated and the lowest one is returned as an update point from each iterate. Thereafter, the remaining calculation is repeat the iterate over and over until convergence and return the converged point as the global minimum.

Nevertheless, the algorithm has been tested on Ackley function where the global minimum locates at $(0,0)$ that $f(0,0) = 0$. And the first 6 iterates of roots and contours is shown in FIG.3 and

the minimum point (-0.00000034,0.00000003) return by algorithm is shown in TABLE.3. The test result shows that the optimization algorithm is robust to local minima and able to achieve the global minimum convergence. The quest to find to the global minimum pays off handsomely.

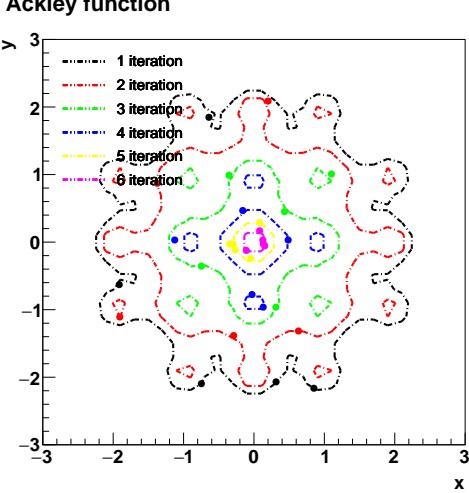

Figure 3: The first 6 iteration results are drawn to illustrate the procedure of optimization test on the Ackley function.

| iteration | average of roots | level of contour |
|:---:|:---:|:---:|
| 0 | (2.00000000,2.00000000) | 6.59359908 |
| 1 | (-0.78076083,-1.34128187) | 5.82036224 |
| 2 | (-0.35105371,-0.62030933) | 4.11933422 |
| 3 | (-0.20087095,0.38105138) | 3.09359564 |
| 4 | (0.06032320,-0.88101860) | 2.17077104 |
| ⋮ | ⋮ | ⋮ |
| 15 | (0.00000404,-0.00000130) | 0.00001199 |
| 16 | (-0.00000194,-0.00000079) | 0.00000591 |
| 17 | (-0.00000034,0.00000003) | |

Table 3: When the optimization method is tested on Ackley function, the average of roots and the level of contour for each iteration is shown above.

In summary, the main procedure of the stochastic contour-based optimization method is decomposed into following steps: 1. Given the initial guess point $x$ for the objective function and calculate the contour level $L$; 2. Solve the equation $f(x) = L$ and get $n$ number of roots. Decompose the set of roots into several convex subsets,return the lowest average of roots as an update point from each iterate; 3. Repeat the above iterate until convergence.

## 5 CONCLUSION

The weak contraction mapping is a self mapping that always map to a subset of domain. Intriguingly, as an extension of Banach fixed-point theorem, the iteration of weak contraction mapping is a Cauchy and yields a unique fixed-point, which fit perfectly with the task of optimization. The global minimum convergence regardless of initial point position and local minima is very significant strength for optimization algorithm. We hope that the advanced optimization with the development of the weak contraction mapping can contribute to empower the modern calculation.

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
