# OpenReview forum: "Weak contraction mapping and optimization"
_ICLR.cc/2019/Conference_

### Official Review · AnonReviewer3 · 2018-10-28
**Ill-defined**

**Rating:** 4
**Confidence:** 5

**Review:**

The authors propose to study what they call weak contraction map. The idea may have its merits, but in the present form, it is not acceptable.

Notably, the key definition of the paper, that is that of the weak contraction mapping (starting with "Then a mapping T : X → X is called weak") is incomplete, because it uses a \mathcal{R}, which is never defined. This makes it hard to evaluate any of the results.

Further, there is no clear separation between text and theorems. Worse, the theorems are not self-contained. E.g. what could perhaps be Theorem 2 (Starting with "x* is a fixed-point of T in X0.") does not define x*.

While I cannot be certain because of the reasons stated above, the authors seem to have had in mind something like the Kakutani theorem:
https://en.wikipedia.org/wiki/Kakutani_fixed-point_theorem
which they don't cite. Their assumptions on the map are weaker than that of Kakutani (upper hemicontinuity), which makes me a bit doubtful as to whether the statements could be proven, even if made precise.

---

> ### Author Response · Authors · 2018-11-06
> **Thank you + reply**
>
>
> Thank you for your reading and comments.
>
> The \mathcal{R} means the range of the map T. And the fixed-point is the point that x* = T(x*).
>
> Thank you for pointing out the part not well defined.
>
> Thank you,

---

### Official Review · AnonReviewer2 · 2018-11-02
**Review: weak contraction mapping and optimization**

**Rating:** 1
**Confidence:** 5

**Review:**

This paper proposes an approach to zeroth order optimization based on the Banach fixed point theorem for contractive maps. They define a "weak contraction map," argue that it will have a unique fixed point, and use this to propose a zeroth order optimization algorithm which iteratively identifies sublevel sets of the objective until convergence to the optimum.

At each iteration $t$, the $f(x_t)$-sublevel set of $f$ is found using a root-finding algorithm, and the next point $x_{t+1}$ is calculated by averaging a collection of points on the boundary of the sublevel set.

My main concern about this paper is that the optimization algorithm works neither in theory nor in realistic practical scenarios. There are two main issues:
(1) Identifying the sublevel sets requires solving equations of the form $f(x) = L$, which is just as hard as optimizing $f$ in the first place! In many realistic scenarios, e.g. machine learning problems, you know what the minimum value of the function is, so you could just solve $f(x) = f^*$ and be done in one step! Even if you don't know the optimal value, you could do some version of binary search. Also, for the two or three dimensional problems with relatively simple expressions that the authors experimented on, finding roots might be possible, but for higher dimensions or more complicated functions, finding these roots would require numerical optimization--which is the problem we are trying to solve in the first place.
(2) The authors seem to imply that this algorithm would work for any $f$, however, consider the function in 1 dimension $f(x) = 1$ for all $x \neq x^*$, and $f(x^*) = 0$. For this function, the $f(x_t)$-sublevel sets are the entire domain until $x_t = x^*$. It is unclear what "points on the contours" would mean in this case, but whatever those contours are, the algorithm would never converge on this function because the function value of $x_0, x_1, ...$ would all be the same, so the contours would remain the same. This function might seem a little ridiculous, but continuous or even Lipschitz version of this counterexample could be constructed by smoothing things out around $x^*$, and a function such as this could be obfuscated by writing it down with a long, complicated expression making it hard to identify $x^*$ by inspection.

There are some typos/typesetting issues. It seems that all of the theorem and lemma statements in Section 2 are missing the bold "Theorem" and "Lemma" heading. In the paragraph after equation (2), "weak contraction mapping" is defined twice, I believe the first definition should be just a "contraction mapping."

---

> ### Author Response · Authors · 2018-11-06
> **Some clarification**
>
> Thank you for your reading and comments.
>
> I have some clarification with regard to the comments:
>
> 1) I can see your point that some root-finding algorithm is based on or related to optimization. While the implementation of root-finding algorithm is not the scope of discussion. In other words, the logic and argument in this paper could be " give there is a reliable root-finding algorithm, it is possible to build a contraction map to find the global optimum". The package I am using for root finding is "Math/MultiRootFinder.h" and "Math/WrappedMultiTF1.h" of ROOT software;
>
> 2) For the function $f(X) = 1$, it is possible build some non-expansive map, but not weak contraction map. There is a requirement for weak contraction map that " D(X_{i+1} < q D(X_{i})$, which guarantee the existence of fixed-point. In other words, it also requires or indicates the objective function has a unique minimum but don's need to be convex.
>
> Best regards,

---

> > ### Comment · AnonReviewer2 · 2018-11-06
> > **clarification**
> >
> > Regarding 1), I understand that your argument is that you can build a weak contraction map in order to find the global optimum IF you have a reliable root-finding algorithm. My point is that, if you have a reliable root-finding algorithm, there is no need to build a weak contraction map in the first place, you can simply find the optimum using the root finder. In many cases, the minimum value is a known value f* (e.g. in machine learning the optimum is typically f*=0) so you can optimize by finding the root f(x) = f*. Even if that is not the case, you can perform some kind of "binary search" on the value using a small number of calls to the root finding algorithm.
> >
> > Regarding 2), the function I suggested was f(x) = 1 for x \neq x* and f(x*) = 0, so there is still a unique minimizer. The issue is that in order to make the weak contraction map that converges to x*, your sets X_1,X_2,... all need to contain x*, and the need to keep zooming in until they contain only x*. However, unless you already know what x* is, I don't see how you could ensure that you will keep zooming in on it. And if you do know what x* is, you don't need to optimize in the first place.

---

> > > ### Author Response · Authors · 2018-11-06
> > > **clarification**
> > >
> > >
> > > 1) In general case, the minimum value is not known value(unless intend to overfit the data) because the degree of freedom of objective function is limited by the number of parameters and the way objective function formed.  The objective function can be very close or like to the data but not exactly the same.
> > >
> > > Yes, if this algorithm is regarded as some kind of "binary search" in high dimensional space. Then the paper is trying to explain the mechanism of the "binary search" and why it can converge to the global minimum as an extension of Banach fixed point theorem especially in the case that the objective function has many minima. Therefore, the novelty of this paper is focused on the extension of Banach fixed point theorem or the variation with respect to Banach fixed point theorem.
> > >
> > > 2)For the function f(x)=1, indeed any x in the domain is a minimizer but it is not a unique minimizer. It is possible to build non-expansive mapping rather than contraction mapping.  Solutions exists for this case but they are not unique.  And this is not the scope of this paper. The weak contraction mapping is proposed that trying to find the global minimum point that the objective function has a unique global minimum but not has to be convex. Ackley function is a good example. The paper shows that it is possible to make the range of the mapping shrink to a point.
> > >
> > > I corrected the typesetting and updated the manuscript.  I hope the updated version can help to clarify some existing misunderstanding.
> > >
> > > Thank you,

---

### Official Review · AnonReviewer1 · 2018-11-21
**The paper is highly unconvincing**

**Rating:** 3
**Confidence:** 2

**Review:**

This paper considers self-maps of metric spaces where the range is strictly smaller than the domain.  Under this condition this tries to show that such a map has a fixed point.  Now the paper suggests that such a "weakly contractive" map has a fixed point and tries to use such maps to find the global minima of functions.

Even if all the proofs in this paper were right I do not see what this has anything to do with learning and why such a paper has been submitted to ICLR! This paper should probably be submitted to an optimization journal!

The basic proofs here are completely unclear. Like in Lemma 1.2, its not even clear what the variable "x" is in the limit! The limit is being taken over the sequence index as far as I can see. Top of page 3 tries to describe an algorithm which can leverage weak contraction to get the global minima if it exists. But this description is hardly making any sense to me. I don't see how the function to be optimized is being used to define the weakly contractive "T" map in the paragraph just below the proof of Lemma 1.4. (How is that parameter "h" even chosen in the definitions of X^{>} and X^{\leq}?)

Without a clear pseudocode there is almost nothing concrete in the paper to judge correctness by. The experiments are all set-up on standardized functions which have nothing to do with learning setups. So the relevance of the experiments is completely unclear, let alone the fact that the description is too muddled up.

Also the notation used in the paper is highly non-standard and that makes reading very difficult. For example "D" seems to be the symbol for diameter of the metric space. So D is a property of the metric space (X,d) and its not a part of the definition of the metric-space as the weird notation "(X,d and D)" seems to suggest!  Also the definition 2 is ambiguous because it uses a {\cal R} which doesnt seem to have been defined anywhere!

---

### Meta-Review · Area_Chair1 · 2018-12-11
**ICLR 2019 decision**

**Confidence:** 5
**Recommendation:** Reject

**Metareview:**

This paper proposes an optimization algorithm based on 'weak contraction mapping'. The paper is written poorly without clear definitions and mathematical rigor. Reviewers doubt both the correctness and the usefulness of the proposed method.  I strongly suggest authors to rewrite the paper addressing all the reviews before submitting to a different venue.